# Cancer Cells Tune the Signaling Pathways to Empower de Novo Synthesis of Nucleotides

**DOI:** 10.3390/cancers11050688

**Published:** 2019-05-17

**Authors:** Elodie Villa, Eunus S. Ali, Umakant Sahu, Issam Ben-Sahra

**Affiliations:** 1Department of Biochemistry and Molecular Genetics, Northwestern University, Chicago, IL 60611, USA; elodie.villa@northwestern.edu (E.V.); eunus.ali@northwestern.edu (E.S.A.); umakant.sahu@northwestern.edu (U.S.); 2Robert H. Lurie Cancer Center, Northwestern University, Chicago, IL 60611, USA

**Keywords:** de novo nucleotide synthesis, oncogenes, tumor suppressors, mTORC1, MYC, RAS, AKT, cancer metabolism, short term and long-term regulation, metabolic vulnerability

## Abstract

Cancer cells exhibit a dynamic metabolic landscape and require a sufficient supply of nucleotides and other macromolecules to grow and proliferate. To meet the metabolic requirements for cell growth, cancer cells must stimulate de novo nucleotide synthesis to obtain adequate nucleotide pools to support nucleic acid and protein synthesis along with energy preservation, signaling activity, glycosylation mechanisms, and cytoskeletal function. Both oncogenes and tumor suppressors have recently been identified as key molecular determinants for de novo nucleotide synthesis that contribute to the maintenance of homeostasis and the proliferation of cancer cells. Inactivation of tumor suppressors such as *TP53* and *LKB1* and hyperactivation of the mTOR pathway and of oncogenes such as *MYC*, *RAS*, and *AKT* have been shown to fuel nucleotide synthesis in tumor cells. The molecular mechanisms by which these signaling hubs influence metabolism, especially the metabolic pathways for nucleotide synthesis, continue to emerge. Here, we focus on the current understanding of the molecular mechanisms by which oncogenes and tumor suppressors modulate nucleotide synthesis in cancer cells and, based on these insights, discuss potential strategies to target cancer cell proliferation.

## 1. Introduction

Signaling systems allow cells to sense their internal and external surroundings in an integrated manner and generate harmonized responses comprising processes such as growth, proliferation, differentiation, and survival. The development of cancer involves consecutive alterations encompassing signaling, metabolic and genetic modifications that that empower cells to escape self-regulating mechanisms that generally eliminate the survival of abnormally proliferating cells. Although cancer cells are heterogeneous in origin and cell type, most cancer cells share features called hallmarks [1]. Some changes are gain-of-function mutations, causing oncogenes that spur tumor formation; others disable tumor suppressor genes that normally prevent cells from growing improperly or surviving outside their usual niche. Tumorigenesis is generally accompanied by cellular metabolic reprogramming that enables cancer cells to adapt to and sustain the energetic demands required to support growth, proliferation, and survival. A well-described mechanism of metabolic alteration observed in cancer cells is the fermentation of glucose to lactate, called the Warburg effect, where the rate of glycolysis is increased in tumor cells, as compared to that in normal cells [2]. This increase in aerobic glycolysis enables the accumulation of metabolic intermediates required for anabolic reactions, increasing the biomass essential for cancer cell growth and proliferation [3].

Mechanisms for proper cell division require the preservation of nucleotide pools used for DNA and RNA synthesis. Nucleotides can be produced through salvage pathways, via the recycling of existing nucleosides and nucleobases, or through the de novo synthesis pathways, using amino acids and small molecules to build the purine and pyrimidine rings. Unlike nonproliferating cells, proliferating cells such as immune cells and cancer cells are predisposed to use the de novo nucleotide synthesis pathways [4,5]. The mechanisms explaining the metabolic shift from a normal to a high rate of de novo nucleotide synthesis in cancer cells involve coordinated inputs from metabolic and signaling pathways [6]. De novo biosynthesis of both purines and pyrimidines has been observed to be altered in cancer and requires the generation of 5-phosphoribose-1-pyrophosphate (PRPP), the activated form of ribose derived from ribose 5-phosphate, which is produced through the oxidative and nonoxidative arms of the pentose phosphate pathway (PPP) parallel to glycolysis. The pyrimidine ring is first assembled from glutamine, bicarbonate, and aspartate and is then attached to PRPP through six reactions. The first three reactions in the de novo pyrimidine synthesis pathway are catalyzed by one cytosolic tricatalytic enzyme called carbamoyl phosphate synthetase 2, aspartate transcarbamylase, and dihydroorotase (CAD), which produces dihydroorotate. Then, dihydroorotate enters mitochondria, where it is oxidized to orotate by dihydroorotate dehydrogenase (DHODH). UMP synthase (UMPS) converts orotate through two catalytic reactions into uridine monophosphate (UMP) (Figure 1A). 

Purine synthesis differs from pyrimidine synthesis in that all reactions occur in the cytosol, and the purine ring is directly built onto the activated ribose, PRPP. The purine ring is synthesized from various substrates, including glutamine, glycine, bicarbonate, and N10-formyl-tetrahydrofolate (THF). After a 10-step reaction, inosine monophosphate (IMP) is produced and converted into guanosine monophosphate (GMP) (via the enzymes inosine monophosphate dehydrogenase (IMPDH) and guanosine monophosphate synthetase (GMPS)) or adenosine monophosphate (AMP) (via metabolic reactions involving adenylosuccinate synthase (ADSS) and adenylosuccinate lyase (ADSL)) (Figure 1B). Once formed, the ribonucleoside triphosphates (NTPs) produced de novo can be utilized for RNA synthesis. However, DNA synthesis requires the cytoplasmic reduction of NTPs to deoxy-NTPs catalyzed by the NAPDH-dependent enzyme ribonucleotide reductase (RNR). Nitrogen and carbon that feed de novo nucleotide synthesis are provided by glutamine, aspartate and several glucose-derived metabolites originating from the PPP, the serine/glycine pathway, and one-carbon metabolism [4] (Figure 1).

It is well recognized that cancer cells rewire their metabolism to enhance de novo nucleotide synthesis in order to grow and proliferate; however, the molecular events by which oncogenes or tumor suppressors modulate these metabolic pathways are not fully elucidated. Here, we systematically review the literature defining the influence of signaling pathways on nucleotide metabolism by first focusing on the short-term molecular features involving posttranslational modifications, then on the long-term processes comprising transcriptional mechanisms and finally on the reemergence of nucleotide metabolism as a new targetable weakness for cancer therapy.

## 2. Acute Regulation of Nucleotide Synthesis by Oncogenes and Tumor Suppressors

Altered expression and activity of metabolic enzymes, including enzymes involved in nucleotide synthesis, are regulated by oncogenes and tumor suppressor genes [7]. To examine the mechanism triggering the alteration of nucleotide synthesis characteristically observed in cancer cells, a better understanding of the acute and direct molecular regulation of nucleotide synthesis pathways by signaling systems is critical to identify the initial metabolic events for eventual therapeutic targeting in cancers.

As mentioned above, de novo nucleotide synthesis is activated in proliferating cells in response to the enhanced demand for nucleotides to support RNA and DNA synthesis. Recently, several studies have indicated that de novo nucleotide synthesis required for cancer cell proliferation is directly regulated by oncogenes and tumor suppressors. An increasing number of oncogenes (e.g., RAS and AKT) and tumor suppressors (e.g., AMP-activated protein kinase (AMPK)) and the primary metabolic sensor (mTOR complex 1 (mTORC1)) described below are being implicated in the acute regulation of nucleotide metabolism [8,9,10,11].

### 2.1. Acute Regulation of Pyrimidine Synthesis by Growth-Promoting Signals

Mechanisms supporting cell growth and anabolic metabolism are coordinated by the mechanistic target of rapamycin (mTOR) [12]. mTORC1 plays an essential role in regulating ribosome biogenesis, glucose metabolism, lipogenesis, nucleotide synthesis, and autophagy. 

mTORC1 is a signaling and metabolic hub that senses nutrients and energy as well as other growth signals to regulate increases or decreases in anabolic and catabolic processes [13,14,15,16]. Growth factor signaling through the activation of the phosphatidylinositol 3-kinase-Akt (PI3K/Akt) pathway stimulates mTORC1 signaling by maintaining the spatial localization of the heterotrimeric TSC1/TSC2/TBC1D7 complex, a guanosine triphosphatase (GTPase)-activating protein complex (GAP), which negatively regulates the activity of the small GTPase Rheb located at the lysosomal surface [17,18,19]. Upon sensing growth signals, the TSC complex disengages from the lysosome and enables an increase in the abundance of Rheb GTP-bound forms, which activate mTORC1 signaling. The PI3K-Akt-mTORC1 signaling pathway has been reported to be dysregulated by various mechanisms in a significant fraction of human tumors [20]. In proliferating cells, mTORC1 was shown to promote the synthesis of pyrimidines through S6K1-mediated phosphorylation of the trifunctional multidomain enzyme CAD on S1859, which catalyzes the first three reactions in de novo pyrimidine synthesis [9,21] (Figure 2). CAD has been reported to bind both Rheb and mTOR, but the metabolic and signaling roles of these regulatory mechanisms remain to be elucidated [22,23]. Furthermore, loss of the tumor suppressor sirtuin 3 (SIRT3) led to an increase in de novo pyrimidine synthesis through the activation of the mTORC1-CAD axis [24]. However, the mechanisms by which SIRT3 loss or inactivation lead to mTORC1 activation remain unknown and may warrant further study.

A study published by Graves and colleagues connected ERK signaling to CAD activity. ERK induces phosphorylation of CAD on Thr456 in the carbamoyl phosphate synthetase 2 (CPS2) domain, which leads to a decrease in the affinity of CAD for the allosteric inhibitor uridine triphosphate (UTP) and a concomitant increase in the binding of the allosteric activator PRPP to the CPS2 domain [8,25]. Two additional phosphorylation sites on CAD have also been reported: Thr1037 (autophosphorylation) and Ser1406; however, the role of these sites in the de novo pyrimidine synthesis activity of CAD remains unknown (Table 1 and Figure 2) [26,27,28]. 

Additionally, Carrey et al., [29] revealed that purified CAD is phosphorylated by cAMP-dependent protein kinase A (PKA) in vitro decreasing the allosteric inhibition of CAD by UTP, a mechanism that should stimulate de novo pyrimidine synthesis. However, a subsequent study showed that PKA phosphorylation caused a decrease in the affinity of CAD for the allosteric activator PRPP [25,30]. Therefore, to unravel this discrepancy, further studies are required to clarify the metabolic role of PKA-mediated CAD phosphorylation in proliferating cells. Additionally, protein kinase C was shown to phosphorylate CAD on Ser1873 and is required for ERK-dependent activation of pyrimidine synthesis [26].

The small GTPase RAS integrates growth signals through the activation of the receptor tyrosine kinase by growth hormones and is therefore involved in the control of cell growth, differentiation and survival [42]. This growth factor-induced signal transduction pathway has long been known to be critical for nucleotide synthesis through its regulation of ribosome biogenesis. Ribosomal biogenesis requires the coordination of protein and nucleotide synthesis to efficiently translate mRNA into proteins. RAS-activated signaling via MEK/ERK/p90RSK drives rRNA synthesis in adult cardiomyocytes, with hypertrophy promoting ribosome biogenesis [43]. Moreover, through its target p90RSK, ERK has been shown to regulate ribosome biogenesis by promoting TIF-1A phosphorylation [44]. 

Multiple lines of evidence indicate that ERK activation leads to long-term stimulation of de novo pyrimidine and purine synthesis through the regulation of the transcription factor c-MYC [33,34]. It is also tempting to speculate that, in addition to the ERK-mediated posttranslational regulation of CAD, the RAS/ERK pathway could directly influence the activity of de novo purine synthesis enzymes to acutely control flux through this metabolic pathway in normal proliferating cells and especially in cancer cells with ERK hyperactivation.

### 2.2. Acute Regulation of PRPP Availability for de Novo Nucleotide Synthesis by Signaling and Metabolic Pathways

To control cell growth and survival, the PI3K/Akt pathway integrates environmental signals, notably growth factor signaling. Saha et al., (2014) demonstrated that upon various growth cues, Akt binds to transketolase (TKT), a key enzyme in the nonoxidative PPP [10] and directly phosphorylates and activates TKT, enhancing carbon flow from glycolytic intermediates towards the non-oxidative PPP, thereby increasing PRPP availability for nucleotide synthesis [10] (Figure 2). 

PRPP is synthesized by the enzyme phosphoribosylpyrophosphate synthetase (PRPS1/2/L1) by a one-step transfer of the β,γ-diphosphoryl group of adenosine triphosphate (ATP) to the C-1 hydroxyl group of α-d-ribose 5-phosphate. The availability of PRPP can be modulated by the allosteric binding of a purine nucleotide (ADP) to PRPS, which modulates nucleotide production in mammalian cells [45]. Recently, the energy sensor AMPK was shown to directly regulate the activity of PRPS1/2 upon metabolic stress. Glucose deprivation resulted in the AMPK-mediated phosphorylation of PRPS1 and PRPS2 on Ser180 and Ser183, respectively, leading to the conversion of PRPS hexamers to monomers, thereby inhibiting PRPS1/2 activity and, consequently, nucleotide synthesis [11] (Figure 2). This report is consistent with another study in which the subcellular localization of the purine enzyme formylglycinamidine ribonucleotide synthase (FGAMS) was found to be sequestered in cytoplasmic granules in response to AMPK activation. Therefore, this AMPK-dependent sequestration of FGAMS from the purine multi-enzyme complex, called purinosome, could be one mechanism by which de novo purine synthesis can be stalled in human cells in response to energy stress [46].

Glycolysis produces ATP and metabolite intermediates that can be immediately used for anabolic metabolism [47]. Pyruvate kinase M2 has been shown to play a critical role in controlling flux through the glycolytic side branch pathways sustaining anabolic metabolism. Glucose metabolism through the oxidative and nonoxidative arms of the PPP is responsible for the production of ribose-5-phosphate, a major intermediate necessary for nucleotide synthesis, NAD, and histidine metabolism [48,49,50]. Studies suggest that the rate-limiting enzyme for pyruvate generation in glycolysis, pyruvate kinase—more specifically, pyruvate kinase M2 (PKM2), an isoform highly expressed in cancer cells—is essential to modulate metabolic flux from glycolysis toward ribose-5-phosphate and serine biosynthesis, thereby promoting de novo nucleotide synthesis [32,51,52,53]. However, the exact mechanisms by which pyruvate kinase regulates nucleotide synthesis remain obscure.

The oxidative branch of the PPP producing NADPH from NADP^+^ provides a pool of reducing equivalents for the synthesis of both nucleotides and lipids. Moreover, NADPH is essential for counterbalancing the high levels of reactive oxygen species (ROS) generated by increased metabolic activity [54]. A recent study by Hoxhaj and colleagues showed that insulin signaling through the activation of the PI3K/Akt pathway induces rapid synthesis of NADP^+^. Akt directly phosphorylates and stimulates NAD kinase (NADK) in mammalian cells, thereby increasing the availability of cellular NADPH, which is required to mitigate stress from high ROS levels and sustain NADPH-dependent anabolic production of nucleotides and lipids in proliferating cells [31,55,56] (Figure 2). 

The basic understanding of the immediate regulation of nucleotide synthesis by multiple signaling systems is emerging and remains to be explored.

## 3. Slow Regulation of Nucleotide Synthesis by Oncogenes and Tumor Suppressors

Nucleotide synthesis is a requirement for the growth and replication of proliferating cells. Oncogenic activation of the growth signaling pathways reprograms cellular metabolism, which may impact nucleotide synthesis pathways. These aberrant signaling pathways activate key transcription factors that stimulate a metabolic gene network program connected to the uptake of nutrients (glucose, glutamine, etc.), the activation of specific metabolic conduits and, more importantly, the synthesis of macromolecules such as proteins, lipids and nucleic acids from metabolic precursors [57,58]. Here, we will discuss the role of oncogenes and tumor suppressors in the regulation of nucleotide synthesis through transcriptional mechanisms.

### 3.1. c-MYC, a Master Regulator of Nucleotide Synthesis in Eukaryotic Cells 

MYC was first identified as a cellular homolog of v-myc, a retroviral gene that was found to induce tumorigenesis in chicken cells [59,60]. The protooncogene c-MYC (MYC) regulates the expression of nearly 15% of the human genome and is upregulated in 50% of human cancers [61,62,63]. In several mouse models, overexpression of MYC leads to tumor growth, while MYC suppression results in decreased tumorigenesis [64,65]. MYC is a 48 kDa DNA-binding transcription factor with an N-terminal transcription activation domain, a C-terminal helix-loop-helix (bHLH) and a leucine zipper (LZ) domain [66]. MYC interacts with Max via the leucine zipper, followed by binding of the bHLH domain to target sequences [67]. MYC primarily regulates genes associated with nucleotide synthesis by directly binding to the promoters of most genes involved in nucleotide metabolism and positively regulating their expression [4,68,69,70] (Figure 3). MYC also regulates the expression of metabolic genes providing metabolite precursors for nucleotide synthesis. Cunningham et al. [35] demonstrated that MYC coordinates the production of proteins and nucleic acids in cancer cells through the upregulation of PRPS2 isoform. During MYC hyperactivation, eIF4E controls *Prps2* mRNA translation through a specialized cis-acting regulatory element and directs an increase in nucleotide biosynthesis. In addition, MYC controls the expression of genes associated with the uptake and catabolism of glutamine required for de novo nucleotide synthesis and other metabolic processes [71]. Hyperactivated MYC is a common feature of many malignancies, and studies have demonstrated that MYC depletion mostly results in the inhibition of purine synthesis and growth suppression in various types of cancer cells [68,72]. For example, Wang et al. [73] demonstrated that MYC enhances de novo purine synthesis in a glioblastoma model. Interestingly, the tumorigenicity of glioblastoma tumor cells was either susceptible to the downregulation of MYC or to the direct inhibition of purine metabolic gene expression. 

Understanding the complexity of MYC-mediated metabolic rewiring in cancers, along with the ways in which MYC cooperates with other signaling drivers such as the mTORC1 and RAS/ERK pathways, can provide translational prospects for cancer therapy.

### 3.2. The RAS-RAF-ERK-MYC Axis Controls Nucleotide Synthesis in Proliferating Cells

MYC is activated downstream of the RAS-ERK pathway [74,75]. K-RAS mutations are frequent in different types of cancer [76], being found in approximately 15% of human cancers [77]. Oncogenic K-RAS reprograms cellular metabolism by promoting glucose uptake, shifting glucose intermediates into the hexosamine biosynthesis and nonoxidative pentose phosphate pathways, and altering glutamine metabolism in tumors [71,78]. Furthermore, oncogenic K-RAS maintains high intracellular nucleotide levels by enhancing de novo synthesis of purines and pyrimidines in pancreatic ductal adenocarcinoma (PDAC) through upregulating MYC-mediated transcriptional activation of ribose 5-phosphate isomerase A (RPIA), a gene involved in the nonoxidative PPP [79] (Table 1). RPIA catalyzes the reversible conversion of ribose-5-phosphate to ribulose-5-phosphate and has been shown to play a crucial role in the development of human hepatocellular carcinoma (HCC) through ERK signaling [79]. Enhanced PPP activity results in elevated de novo nucleotide synthesis, which is required for actively proliferating cancer cells [79].

Downstream of K-RAS, the oncogenic activation of BRAF, which encodes a serine-threonine kinase, also maintains nucleotide synthesis through MYC activation. BRAF mutations are common in various human cancers; the point mutation V600E accounts for 80% of all BRAF mutations, especially in melanoma [77]. BRAF promotes the transcription of a melanoma gene signature in embryonic neural crest progenitors, which later develop into tumors. In addition, DHODH inhibition prevents the transcription of genes required for neural crest development and melanoma growth in zebrafish models [80]. Thus, simultaneous pharmacological inhibition of DHODH and BRAF has been reported to decrease the progression of melanoma in cell lines and in mouse xenograft models. Oncogenic BRAF inhibits senescence and apoptosis and promotes angiogenesis, tissue invasion and metastasis [81]. The contribution of BRAF-induced metabolic alterations in these metastatic processes is under investigation.

### 3.3. Oncogenic Activation of RAS and Loss of Tumor Suppressors Reprogram Nucleotide Metabolism

Liver kinase B1 (LKB1) is a serine/threonine kinase inactivated in a range of cancers. Interestingly, inactivation of LKB1 is often accompanied by mutations in the RAS-RAF pathway in human cancers [82]. A study showed that concurrent loss of LKB1 and activation of KRAS stimulates flux into the serine, glycine, one-carbon (SGOC) metabolic network leading to enhanced S-adenosyl-methionine (SAM) synthesis and tumorigenesis in PDAC [83]. This study linked dysregulated serine metabolism to alterations in the epigenetic landscape as a plausible cause of many cancers. Moreover, simultaneous activation of KRAS and loss of the tumor suppressor LKB1 modulates the urea cycle to promote de novo pyrimidine synthesis. In lung cancer cells, loss of LKB1 and activation of K-RAS led to increased expression of mitochondrial carbamoyl phosphate synthetase-1 (CPS1), increasing the abundance of carbamoyl phosphate intermediates that can exit the mitochondria to be utilized by the de novo pyrimidine synthesis pathways [41]. Interestingly, crosstalk between pyrimidine synthesis and the urea cycle was also detected in a subset of cancer cells with low argininosuccinate synthase (ASS1) expression. ASS1 is important for the detoxification of ammonia, as it catalyzes the synthesis of arginosuccinate from citrulline and aspartate, an essential step in urea synthesis [84,85] (Figure 3). The downregulated ASS1 expression in these cancer cells increases the aspartate pool, which can be channeled to CAD for the synthesis of pyrimidines.

The tumor suppressor protein 53 (TP53 or p53) is a transcription factor that regulates the expression of genes with diverse cellular functions. p53 is a major tumor suppressor found to be mutated in more than 50% of human cancers [86]. Accumulating evidence suggests that the activated form of mutant p53 (mtp53) reprograms cellular metabolism to sustain growth and proliferation. p53 knockdown results in reduced ribose and deoxyribose nucleotide pools, which in turn inhibits the proliferation of several breast cancer cell lines [38]. Expression of mtp53 enhances the expression of several nucleotide metabolism genes, such as deoxycytidine kinase (DCK), thymidine kinase 1 (TK1), thymidylate synthetase (TYMS), IMPDH1, IMPDH2, GMPS, dihydrofolate reductase (DHFR), and RNR (RRM1 and RRM2). Thus, p53 regulates cellular signaling and regulates the invasive potential of tumor cells via nucleotide metabolism. The p53 mutant breast and pancreatic cancer cells exhibit augmented sensitivity to treatment with gemcitabine, a well-known chemotherapeutic drug activated upon phosphorylation by DCK [87].

### 3.4. Transcriptional Control of de Novo Nucleotide Synthesis through mTORC1 Signaling in Proliferating Cells

Unlike de novo pyrimidine synthesis, de novo purine synthesis is stimulated by mTORC1 signaling in a more delayed manner via the regulation of the transcription factors MYC, sterol regulatory element binding protein (SREBP) and activating transcription factor 4 (ATF4), which induce the expression of specific metabolic enzymes required for de novo synthesis of purines. For example, ATF4 downstream of mTORC1 signaling, stimulates gene expression of the serine/glycine synthesis pathway and the mitochondrial tetrahydrofolate cycle, which produces the one-carbon formyl units necessary for purine ring assembly in the cytosol [36] (Figure 3). More specifically, in response to growth signals, activation of the mTORC1-ATF4 axis induces an increase in the transcription of a gene called MTHFD2. MTHFD2 is a mitochondrial NAD+-dependent 5,10-methylene-THF dehydrogenase and is essential for the synthesis of mitochondria-derived formate, which is required for purine ring assembly [88]. Moreover, a recent study demonstrated that downregulation of protein kinase C λ/ι (PKC λ/ι) in neuroendocrine prostate cancer cells stimulates mTORC1 signaling, leading to ATF4-dependent upregulation of the SGOC pathway. Depletion of ATF4 or pharmacological inhibition of mTORC1 signaling reduced the expression of genes associated with the SGOC pathway and, therefore, tumorigenesis [89]. Furthermore, via unidentified mechanisms, mTORC1 has been proposed to stimulate the formation and assembly on the mitochondrial surface of the purinosome, which carries out de novo purine synthesis [90]. The formation of this complex of proteins is believed to generate flux through this metabolic pathway [91].

The proper execution of the regulated nucleotide synthesis program downstream of mTORC1 needs substantial input of nutrients and energy. Therefore, the activation state of mTORC1 is robustly influenced by the availability of building blocks (amino acids, glucose, cholesterol, and nucleotides) for the enhanced synthesis of macromolecules (proteins, lipids, and nucleic acids) [92,93,94,95]. Recent studies have demonstrated that mTORC1 can sense the intracellular levels of purines but not pyrimidines through the TSC complex [94,95], suggesting that pyrimidines are sensed through mechanisms independent of mTORC1 signaling. The role of mTORC1 in the control of macromolecular synthesis and its position on the surface of the lysosome raise the question of whether the lysosome can recycle macromolecules to convey signals to mTORC1 signaling. Upon mTORC1 inhibition, Nuclear fragile X mental retardation-interacting protein 1 (NUFIP1) is transported from the nucleus to lysosomes and more importantly to autophagosomes stimulating the degradation and recycling of ribosomes, a process called ribophagy [96].

Upstream of mTORC1, activation of the PI3K/Akt pathway has been shown to stimulate nucleotide synthesis. Interestingly, loss of the tumor suppressor PTEN, a lipid phosphatase that dephosphorylates phosphatidylinositol (3,4,5)-triphosphate (PIP3) to phosphatidylinositol (4,5)-bisphosphate (PIP2), results in the activation of AKT and mTORC1 signaling and thus the promotion of cell growth and proliferation and the inhibition of apoptosis [97,98]. PTEN is disrupted in a wide range of human cancers [99]. A recent study established that loss of PTEN enhances the growth of mouse embryonic fibroblasts (MEFs) in a glutamine-dependent manner, enhancing the flux of metabolites through the de novo pyrimidine synthesis pathway [37]. Because mTORC1 activity is increased in response to PTEN loss, flux through the de novo pyrimidine synthesis pathway is also augmented in this setting through CAD activation [9]. In addition, pharmacological inhibition of DHODH selectively reduced the growth of PTEN-null cells, supporting the nucleotide synthesis dependency hypothesis in cells with constitutive activation of the PI3K-AKT-mTOR pathway.

### 3.5. Indirect Regulation of de Novo Nucleotide Synthesis by the Hippo-Yap Pathway

The Hippo signaling pathway regulates organ size and tissue growth in response to surrounding signals [100]. These exogenous signals influence the activity of the Hippo kinase cascade, thereby regulating the nuclear localization of the transcriptional coactivators YAP and TAZ, which bind to the TEAD family of transcription factors to stimulate tissue growth [101]. Activated Yap1 stimulates the development of hepatomegaly and potentiates tumor formation in the zebrafish liver [39]. Moreover, Yap1 induces the expression of glutamine synthetase (GLUL), which increases glutamine levels and drives de novo nucleotide synthesis. Inhibition of GLUL diminishes flux through nucleotide synthesis pathways, thereby decreasing hepatomegaly and liver cancer, indicating that Yap1-driven liver tumorigenesis is vulnerable to the inhibition of nucleotide synthesis. The findings in this study are consistent with those of a recent study demonstrating that YAP1, via the induction of the glucose importer GLUT1, enhances glucose uptake and utilization to stimulate de novo nucleotide synthesis. *YAP1* depletion resulted in reduced organ growth, and this phenotype was partially rescued by providing exogenous nucleosides [40]. The metabolic role of the Hippo-YAP-TAZ pathway in the regulation of nucleotide metabolism merits further attention, as the specific mechanisms involving the differential regulation of pyrimidine and purine synthesis by YAP or TAZ remains mostly unknown.

All of these findings highlight the existence of lines of communication connecting tumor suppressors, oncogenes and nucleotide synthesis in cancers. However, understanding whether these signaling and metabolic routes could be used to define potential therapeutic avenues to target specific cancers requires further studies.

## 4. Nucleotide Synthesis Is Reemerging as a Metabolic Vulnerability in Cancer

The examination of the distinct metabolic dependencies of many tumors provided potential metabolic targets for exploiting nucleotide metabolism, some of which are being evaluated in preclinical models and clinical trials. Here, we discuss past and current developments in nucleotide metabolism research that have identified metabolic targets and highlighted features that might be exploited to improve cancer therapy.

### 4.1. Antimetabolites for Targeting Metabolic Dependencies in Cancer Cells

The primary medical success of antifolates led to the production of an entire class of drugs known as antimetabolites. Antimetabolites are small molecules that are chemically similar to nucleotide metabolites, but often competitively inhibit the activity of enzymes involved in nucleotide synthesis. Sidney Farber was the first to discover that aminopterin could induce disease remission of some cancers [102]. Aminopterin is the ancestor of the presently extensively used drugs methotrexate and pemetrexed, both of which are folate analogs that inhibit DHFR, which is required for de novo nucleotide synthesis and other metabolic processes [103]. For example, the purine analog 6-mercaptopurine (6-MP) inhibit hypoxanthine-guanine phosphoribosyltransferase (HPRT), a key enzyme in the purine salvage pathway as well as PRPP amidotransferase (PPAT), the first enzyme in de novo purine biosynthesis. 6-MP has been effective in eradicating many cancers [104]. 5-fluorouracil (5-FU), a pyrimidine analog, is structurally similar to uracil but inhibits thymidylate synthase, decreasing the availability of deoxythymidine nucleotides for DNA synthesis. 5-FU and its derivatives are widely used chemotherapeutics against various cancers [105,106]. Other antimetabolite nucleoside analogs, such as gemcitabine and cytarabine, are incorporated into DNA, resulting in the inhibition of DNA polymerases and are frequently used to treat select cancers [107,108]. Cancer cells constantly require nucleotides, which are supplied by enhanced de novo purine and pyrimidine synthesis, not only as a consequence of increased proliferation but also as an adaptation to chemotherapy. Aird et al. demonstrated that cancer cells must maintain a pool of dNTPs for DNA replication and repair [109]. In addition to de novo nucleotide synthesis, it is worth stating that cells can use the nucleotide salvage pathways that enable the recycling of purine and pyrimidine nucleobases and nucleosides. It is tempting to speculate that, in addition to oxygen and nutrient, a tumor also requires nucleosides provided exogenously to survive and proliferate. A recent study from Halbrook and colleagues showed that tumor associated macrophages can release pyrimidines in the microenvironment rendering pancreatic cancer cells resistant to the antimetabolite gemcitabine, which is typically used as a chemotherapeutic agent in various cancers [110]. Interestingly, the inhibition of DHODH in a leukemia mouse model demonstrated that uridine supplementation, through the salvage pathway, could rescue the anti-leukemic effects induced by DHODH inhibition [111]. Furthermore, radiotherapy inducing DNA damage and genotoxic chemotherapies can prime cells to inhibitors of nucleotide synthesis [112,113]. The clinical effectiveness of antimetabolite chemotherapies suggests that the metabolic therapeutic window outspreads beyond the cell proliferation rate and genotoxic response. While the precise molecular mechanisms underlying the distinct efficacy of present antimetabolite therapies are unclear, a better understanding of these and other metabolic therapeutic perspectives may provide indications for the development of more effective and selective cancer treatments.

### 4.2. Emerging Strategies for Targeting Pyrimidine Metabolism in Cancer Cells

Recent studies have revisited the use of nucleotide synthesis inhibitors to target tumor cell progression. 

Targeting cancer metabolism through the inhibition of DHODH has recently received much consideration [114]. Steady-state metabolite profiling in triple negative breast cancer (TNBC) cells treated with prolonged exposure to doxorubicin revealed an increase in the levels of pyrimidine intermediates [112]. Interestingly, under doxorubicin treatment, CAD is phosphorylated on Thr456 by ERK signaling and becomes more sensitive to activation by PRPP (previously described in the acute regulation section, [25]), thus increasing flux through the de novo pyrimidine synthesis pathway. Unsurprisingly, knockdown of DHODH, as well as treatment with DHODH inhibitors (brequinar and leflunomide), sensitized TNBC cells to doxorubicin-induced cell death. DHODH dependency has also been reported in melanoma harboring the BRAF V600E mutation [80], acute myeloid leukemia [111], PTEN mutant cells [37], and K-RAS mutant pancreatic tumor cells [115]. A beguiling question is why cancer cells seem to rely mainly on DHODH and not on CAD, which catalyzes half of the reactions in the de novo pyrimidine synthesis pathway.

Interestingly, DHODH is the only enzyme in the de novo pyrimidine synthesis pathway that is localized in mitochondria and requires oxidative phosphorylation (OXPHOS) to function correctly [116]. Furthermore, it was recently discovered that OXPHOS is required for tumor growth mainly to allow the activation of DHODH by enabling the redox cycling of coenzyme Q, which is essential for maintaining functional de novo pyrimidine synthesis and overcoming cell cycle arrest, thereby promoting tumor growth [117]. In addition, DHODH is an essential enzyme for the function of normal cells, an observation that raises the question of whether targeting DHODH in cancer patients could be a valid therapeutic opportunity. Phase 1 clinical trials using Brequinar did not show any clinical efficacy in cancer patients when this drug was used as a single agent or in combination with cisplatin [118], suggesting that compensation mechanisms exist and could be mediated by an increase in the activity of the pyrimidine salvage pathway. DHODH expression and activity are not linked to an oncogene or tumor suppressor; thus, this enzyme is a nonspecific target, and inhibiting it in cancer patients could cause serious side effects [118]. 

### 4.3. Emerging Strategies for Targeting Purine Metabolism in Cancer Cells

Given that mTORC1 is a master regulator of cellular metabolism and induces metabolic stimulation of de novo purine and pyrimidine synthesis [9,36] in cancer cells, targeting mTORC1 signaling in cancer seems appealing, but rapamycin and rapalogs exhibit only a cytostatic rather than a cytotoxic effect and eventually enable tumor relapse through inducing resistance mechanisms. To identify therapeutic alternatives, exploiting the dependency of mTORC1 on metabolism, especially nucleotide metabolism, has been mechanistically explored [14]. Inhibition of IMPDH, the rate-limiting enzyme in de novo guanylate synthesis, selectively triggered the apoptosis of cells exhibiting growth factor-independent activation of mTORC1 signaling [14]. The antiproliferative effects of IMPDH inhibition in mTORC1-hyperactivated settings stem from the metabolic imbalance arising from the dependency of mTORC1-driven cells on the nucleotide synthesis pathways to maintain ribosomal RNA and DNA synthesis. Furthermore, guanylate depletion induced by IMPDH inhibition does not deactivate mTORC1 signaling, thereby misleading the proliferative cells to keep building macromolecules, eventually triggering DNA replication stress and cell death because of nucleotide insufficiency [14,119]. 

Purine dependency was further observed in acute lymphoblastic leukemia and glioblastoma [73]. In addition, IMPDH addiction was characterized in a subset of small cell lung cancers defined by a low expression level of the transcription factor ASCL1 and high expression of oncogenic MYC [120]. Inhibition of IMPDH with mycophenolic acid (MPA) suppressed the growth of these cancer cells. The results of these two studies highlight the need for high rates of de novo guanylate synthesis and ribosomal biogenesis in tumors. Pharmacological inhibition of purine synthesis as a means to treat cancer has been examined for many years, with IMPDH considered as a possible target. Both MPA and mizoribine are already used in humans as immunosuppressants in organ transplantation and autoimmune diseases; mizoribine is particularly well-tolerated [121,122,123]. Several epidemiological studies have consistently observed an increased risk of cancers in patients receiving organ transplant treated with immunosuppressive drugs [124]. This raises the question of the role of these immunosuppressive drugs in the context of cancer immunotherapies. Cancer immunotherapy is concentrated on the immune system and is frequently more targeted than conventional cancer treatments such as chemotherapy or radiotherapy. The use of chemotherapy based on antimetabolite targeting nucleotide metabolism should certainly not be combined with immunotherapy which is supposedly used to activate the immune system. Further studies are required to estimate the actual cancer risk of the immunosuppressive drugs and notably of IMPDH inhibitors.

## 5. Conclusion and Prospects

The control of cellular metabolism by growth factor-initiated signaling pathways is frequently deregulated in cancer. These oncogenic alterations promote increased nutrient uptake and anabolic metabolism to assemble macromolecules such as the proteins, lipids and nucleic acids indispensable for cell growth. These signaling pathways impact nucleotide metabolism, but the molecular links between these networks and the nucleotide synthesis pathways continue to be explored. The direct roles of mTOR and MYC in the regulation of nucleotide synthesis are currently recognized to be prominent across different types of tumors. Recent studies targeting DHODH or IMPDH reveal metabolic vulnerabilities potentially exploitable in cancer treatment. However, the use of brequinar (a DHODH inhibitor) as a single agent has not yet yielded significant clinical results [114]. One potential strategy would be to combine low doses of metabolic inhibitors with drugs targeting oncogenic signaling pathways to more precisely eliminate the malignancy; undeniably, however, the design of combination therapies with efficacy across many tumor types is challenging. Interestingly, IMPDH inhibitors synergize specifically with different chemotherapies depending on the cancer cell type and determined by specific oncogenic signals [125]. The dependency of the malignancy on these emerging metabolic targets is not always universal. Therefore, blindly targeting metabolism supporting cellular proliferation may not offer the most tolerable therapeutic window, as many nonmalignant cells, as well as cells in the bone marrow, hair follicles, and intestinal crypts, are rapidly dividing. 

Additionally, the cell proliferation rates of normal proliferating cells are often higher than those of cancer cells, and the involuntarily obliteration of normal proliferating cells with antimetabolite chemotherapy can cause serious side effects in patients. Despite these toxicities, antimetabolites are often approved in many modern chemotherapy regimens since they increase the patient survival rate and, in some cases, help eradicate the disease; however, the specificity of these therapies needs to be improved. The signaling and metabolic features of different types of cancers should be considered in the context of cancer therapies, but should be mechanistically investigated to determine effective future treatments specifically targeting cancer cell metabolism influenced by the dysregulation of particular signaling pathways. 

## Figures and Tables

**Figure 1 cancers-11-00688-f001:**
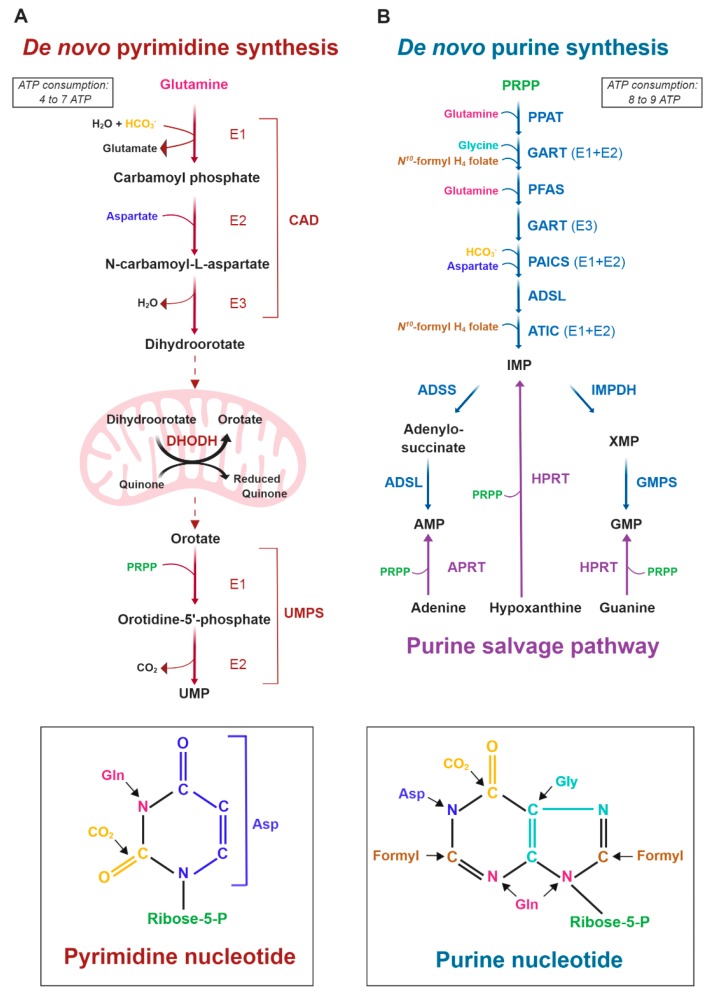
The de novo pyrimidine and purine synthesis pathways. (**A**) Schematic of the de novo pyrimidine synthesis pathway. Pyrimidine synthesis enzymes: CAD: Carbamoyl-Phosphate Synthetase 2, Aspartate Transcarbamylase, And Dihydroorotase; DHODH: Dihydroorotate Dehydrogenase; UMPS: Uridine Monophosphate Synthetase. (**B**) Schematic of the de novo and purine salvage pathways. Purine synthesis enzymes: PPAT: phosphoribosyl pyrophosphate amidotransferase; GART: Glycinamide Ribonucleotide Transformylase; PFAS: Phosphoribosylformylglycinamidine Synthase; PAICS: Phosphoribosylaminoimidazole Carboxylase And Phosphoribosylamino-imidazolesuccinocarboxamide Synthase; ADSL: Adenylosuccinate Lyase; ATIC: 5-Aminoimidazole-4-Carboxamide Ribonucleotide Formyltransferase; IMPDH: Inosine Monophosphate Dehydrogenase; GMPS: Guanine Monophosphate Synthase; ADSS: Adenylosuccinate Synthase; HPRT: hypoxanthine phosphoribosyltransferase; APRT: adenine phosphoribosyltransferase.

**Figure 2 cancers-11-00688-f002:**
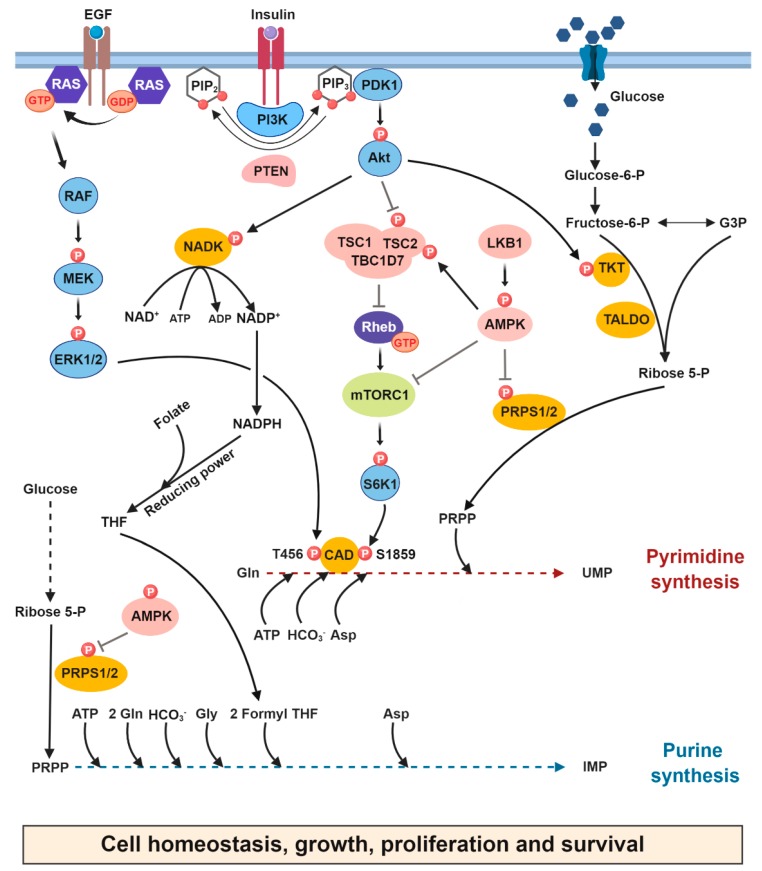
Acute regulation of de novo nucleotide synthesis by oncogenes and tumor suppressors. This schematic highlights our current understanding of how the oncogenic and tumor suppressor signals acutely modulate de novo nucleotide synthesis in cancer cells. In response to growth signals and activated oncogenic RAS, ERK directly phosphorylates CAD on T456 and stimulates de novo pyrimidine synthesis. In addition, mTORC1 activation, downstream of PI3K/Akt signaling, leads to S6K1 mediated-phosphorylation of CAD on S1859, thereby acutely enhancing flux through de novo pyrimidine synthesis. Upstream of mTORC1, Akt phosphorylates TKT on Thr382 and enhances PRPP availability for nucleotide synthesis. Moreover, Akt increases NADP^+^ synthesis through the direct phosphorylation of NAD kinase on S44/S46, thereby increasing the availability of cellular NADPH to sustain NADPH-dependent anabolic production of purine nucleotides. Under metabolic stress, AMPK is activated and directly phosphorylates PRPS1/2 on S180/S183, which inhibits the conversion of ribose 5-phosphate into PRPP reducing its availability for nucleotide synthesis. Tumor suppressors are shown in pink, and key signaling kinases involved in light blue. Metabolic enzymes are shown in orange and small GTPases in dark blue. PRPP, 5’phosphoribosyl-pyrophosphate; PRPS, phosphoribosyl pyrophosphate synthetase; TKT, transketolase; AMPK, AMP-activated protein kinase.

**Figure 3 cancers-11-00688-f003:**
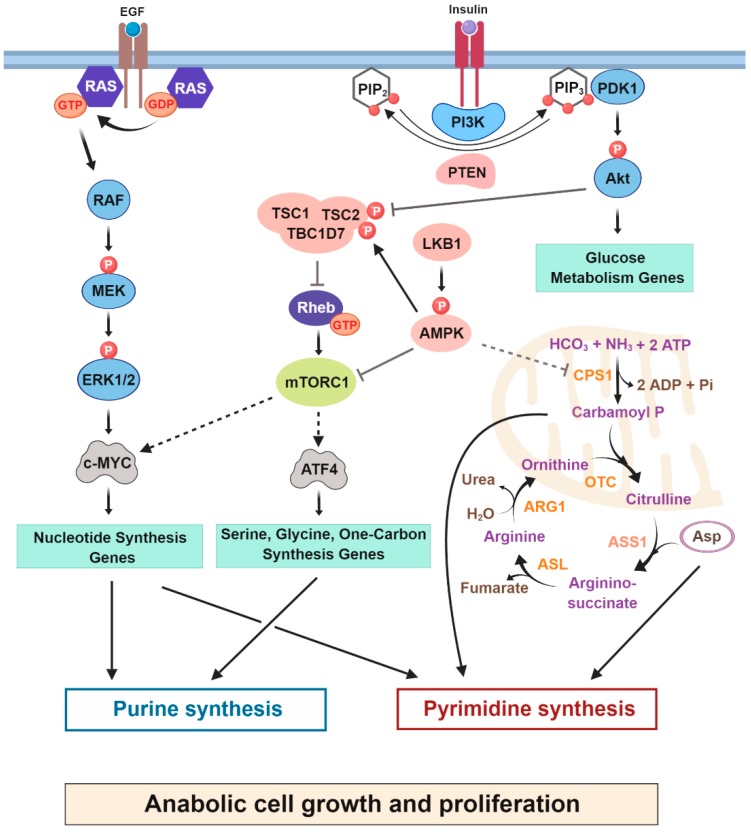
Slow regulation of nucleotide synthesis by oncogenes and tumor suppressors. Growth factor and oncogenic signaling reprogram nucleotide metabolism to increase the biomass essential for cell proliferation. Signaling downstream from PI3K/Akt enhances glucose uptake and glycolysis. The MYC transcription factor, activated downstream from RAS, enhances de novo purine and pyrimidine synthesis. In addition to MYC activation, mTORC1 increases the levels of ATF4 stimulating synthesis of serine, glycine and one carbon formyl units contributing to purine synthesis. Loss of LKB1 and oncogenic activation of K-RAS lead to an increase in CPS1, a key urea cycle enzyme that produces carbamoyl phosphate intermediate supplying de novo pyrimidine synthesis. Tumor suppressors are shown in pink, and key oncogenic signaling kinases in light blue. Metabolic enzymes are shown in orange and small GTPases in dark blue. CPS1- Carbamoyl phosphate synthetase 1, OTC—Ornithine transcarbamoylase, ASS1—Argininosuccinate synthetase 1, ASL—Argininosuccinate lyase, ARG1—Arginase 1.

**Table 1 cancers-11-00688-t001:** Oncogenes and tumor suppressors involved in the short-term and long-term regulation of de novo nucleotide synthesis in tumor cells.

Nature of the Regulation	Regulator(s)	Description of the Molecular Mechanism(s)	References
**Short-term**	**RAS/ERK**	ERK directly phosphorylates CAD on T456 and stimulates CAD activity	[8]
	**PI3K/Akt/** **mTORC1**	mTORC1, through S6K1-mediated phosphorylation of CAD on S1859, enhances flux through pyrimidine synthesis	[9,21]
		Akt mediated-phosphorylation of TKT on Thr382 enhances PRPP availability for nucleotide synthesis	[10]
		Akt phosphorylates NADK on S44/S46 to stimulate the production of NADP(H), an essential cofactor for nucleotide synthesis	[31]
	**SIRT3**	Inactivation of SIRT3 promotes glutamine-dependent de novo nucleotide synthesis in part through hyperactivation of mTORC1 signaling	[24]
	**PKM1**	PKM1 expression impairs nucleotide production and the ability to synthesize DNA and progress through the cell cycle	[32]
**Long-term**	**K-RAS**	Oncogenic K-RAS stimulates nucleotide synthesis through regulation of RPIA expression by c-MYC	[33,34]
	**MYC-eIF4E**	During MYC-driven tumorigenesis, eIF4E controls *PRPS2* mRNA translation through a cis-acting regulatory element and increases nucleotide synthesis.	[35]
	**mTORC1**	mTORC1 signaling, through activation of ATF4, stimulates the expression of MTHFD2 required for one carbon formyl unit incorporation into the purine ring	[36]
	**PTEN**	Loss of PTEN stimulates de novo pyrimidine synthesis through activation of mTORC1 signaling	[37]
	**p53**	Mutant p53 enhances the expression of nucleotide metabolism genes	[38]
	**YAP1**	YAP1 fuels de novo nucleotide synthesis via the stimulation of glutamine synthetase expression (GLUL)	[39]
		YAP1 fuels de novo nucleotide synthesis via the stimulation of glucose transporter 1 expression (GLUT1)	[40]
	**K-RAS and LKB1**	Simultaneous activation of KRAS and loss of LKB1 stimulates de novo pyrimidine synthesis by elevating the expression of carbamoyl phosphate synthetase 1 (CPS1)	[41]

Abbreviations: ERK, extracellular signal–regulated kinase; SIRT3, NAD-dependent deacetylase sirtuin-3, mitochondrial; RPIA, ribose 5-phosphate isomerase A; eIF4E, Eukaryotic Translation Initiation Factor 4E; PRPS2, phosphoribosyl-pyrophosphate synthetase 2; MTHFD2, methylene tetrahydrofolate dehydrogenase 2; CPS1, carbamoyl phosphate synthetase-1; GLUL, glutamine synthetase; TKT, transketolase; PKM1, pyruvate kinase M1.

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
