# Peer review of "Cancer Cells Tune the Signaling Pathways to Empower de Novo Synthesis of Nucleotides"

_cancers, 2019, doi:10.3390/cancers11050688_

Round 1

Reviewer 1 Report

Cancer cells exhibit a dynamic metabolic landscape and require a sufficient supply of nucleotides and other macromolecules to grow and proliferate. The cancer metabolism field mainly focused on central carbon metabolism and less on the cancer-dependent nucleotide biosynthesis. In this review Villa et al. review summarizes well our current understanding of nucleotide biosynthesis in tumors. This review segregates between the acute and long-term regulation of nucleotide biosynthesis. It finished with an overview of the role of nucleotide biosynthesis as a target for cancer therapy.   This is a very helpful review that summarizes the recent work on the nucleotide biosynthesis “do novo” pathway in cancer.

Comments

1.    In row 182 they introduce the term “purinosome” however they elaborate on its definition only on row 328. The definition should be on its first introduction.

2.    The sentence in row 197 (regarding the role of NADPH as a cofactor in cancer) needs to be referenced.

3.    Some of the sentences are written in a different font, (e.g. row 130, 156)

Author Response

Cancer cells exhibit a dynamic metabolic landscape and require a sufficient supply of nucleotides and other macromolecules to grow and proliferate. The cancer metabolism field mainly focused on central carbon metabolism and less on the cancer-dependent nucleotide biosynthesis. In this review Villa et al. review summarizes well our current understanding of nucleotide biosynthesis in tumors. This review segregates between the acute and long-term regulation of nucleotide biosynthesis. It finished with an overview of the role of nucleotide biosynthesis as a target for cancer therapy.   This is a very helpful review that summarizes the recent work on the nucleotide biosynthesis “do novo” pathway in cancer.

Point 1:    In row 182 they introduce the term “purinosome” however they elaborate on its definition only on row 328. The definition should be on its first introduction.

Response 1:  We thank the reviewer for the very nice comment. We have amended the text and added a short definition of purinosome on row 183.

Point 2: The sentence in row 197 (regarding the role of NADPH as a cofactor in cancer) needs to be referenced.

Response 2: Thank you for this comment. We added two references. One from Dr. Diaz-Munoz group (PMID: 29599747) and another from Drs. Metallo and Vander Heiden review (PMID: 23395269).

Point 3:   Some of the sentences are written in a different font, (e.g. row 130, 156)

Response 3: We apologize for this. We wrote the entire review with a Times New Roman font, but after submission, the formatting from the journal must have created some font discrepancies. We have corrected the font accordingly.

Reviewer 2 Report

Summary:

Villa et al contribute a very useful and comprehensive review of recent literature on oncogenic regulation of nucleotide metabolism, including sections for how commonly mutated tumor suppressor/oncogenes post-translationally and transcriptionally regulate nucleotide metabolism. The authors then conclude with a short perspective on how to utilize this information to improve therapies already used to therapeutically in cancer. Below are one major point and minor points to improve the manuscript.

Major comments:

1)     The authors focus on de novo nucleotide synthesis in the review, but given the discussion on antimetabolites, it may be helpful for the authors to discuss other sources of nucleotides for cancer cells. For instance, the uptake of nucleobases and nucleosides from the environment. Many of the antimetabolites discussed are taken up this way, and perhaps nucleobase/side scavenging contributes not only to nucleotide homeostasis in cancer cells but also to the therapeutic window of these drugs. There are also new interesting examples of how alterations in environmental nucleosides affect the efficacy of these drugs (PMID: 30827862) and environmental uridine can rescue the effects of DHODH inhibition (PMID: 27641501). Lastly, ribophagy was recently proposed to provide a short term pool of nucleotides for cancer cells to survive transient periods of starvartion (PMID: 29700228), and this may also be pertinent to discuss as well.

Minor comments:

1)     Lines 271-281 the authors discuss DHODH inhibition in MYC driven cancers. Previously, the authors discuss how MYC primarily drives purine synthesis, while DHODHi will inhibit pyrimidine synthesis only. It would be helpful if the authors could discuss here why inhibiting the pathway MYC does not turn on, actually slows the growth of these cancer cells. For example, would IMPHDi inhibitors actually work better?

2)     With regard to reference 78 (lines 293-298), it may also be helpful to cite PMID: 30100185. This paper provides further evidence that ASS1 loss drives pyrimidine synthesis, and also has a nice discussion on nucleotide imbalance, which the authors later discuss.

3)     In lines 486-489 in discussing the use of IMPDH inhibitors in cancer, it may be worth discussing how the immunosuppressive effects of these drugs will interact with immunotherapies that are increasingly used clinically. How will this affect the ability to target nucleotide metabolism in cancer?

Author Response

Villa et al contribute a very useful and comprehensive review of recent literature on oncogenic regulation of nucleotide metabolism, including sections for how commonly mutated tumor suppressor/oncogenes post-translationally and transcriptionally regulate nucleotide metabolism. The authors then conclude with a short perspective on how to utilize this information to improve therapies already used to therapeutically in cancer. Below are one major point and minor points to improve the manuscript.

Major comments:

1)     The authors focus on de novo nucleotide synthesis in the review, but given the discussion on antimetabolites, it may be helpful for the authors to discuss other sources of nucleotides for cancer cells. For instance, the uptake of nucleobases and nucleosides from the environment. Many of the antimetabolites discussed are taken up this way, and perhaps nucleobase/side scavenging contributes not only to nucleotide homeostasis in cancer cells but also to the therapeutic window of these drugs. There are also new interesting examples of how alterations in environmental nucleosides affect the efficacy of these drugs (PMID: 30827862) and environmental uridine can rescue the effects of DHODH inhibition (PMID: 27641501). Lastly, ribophagy was recently proposed to provide a short term pool of nucleotides for cancer cells to survive transient periods of starvartion (PMID: 29700228), and this may also be pertinent to discuss as well.

Response 1: We thank the reviewer for this critical comment. We focused this review on the regulation of de novo nucleotide synthesis by signaling pathways in cancer because so far the nucleotide salvage pathways have not been shown to be regulated by signaling systems. Nevertheless, we agree with the reviewer and decided to include a little paragraph about the salvage pathways and to cite the three important papers suggested by the reviewer.

Lines 429-438: “In addition to de novo nucleotide synthesis, it is worth stating that cells can use the nucleotide salvage pathways that enable the recycling of purine and pyrimidine nucleobases and nucleosides. It is tempting to speculate that, in addition to oxygen and nutrient, a tumor also requires nucleosides provided exogenously to survive and proliferate. A recent study from Halbrook and colleagues showed that tumor associated macrophages can release pyrimidines in the microenvironment rendering pancreatic cancer cells resistant to the antimetabolite gemcitabine, which is typically used as a chemotherapeutic agent in various cancers PMID: 30827862. Interestingly, inhibition of DHODH in a leukemia mouse model demonstrated that uridine supplementation, through the salvage pathway, could rescue the anti-leukemic effects induced by DHODH inhibition PMID: 27641501.”

Lines 338-345: Recent studies demonstrated that mTORC1 can sense the intracellular levels of purines but not pyrimidines through the TSC complex [91, 92], suggesting that pyrimidines are sensed through mechanisms independent of mTORC1 signaling. The role of mTORC1 in the control of macromolecular synthesis and its position on the surface of the lysosome raise the question of whether the lysosome can recycle macromolecules to convey signals to mTORC1 signaling. Upon mTORC1 inhibition, Nuclear fragile X mental retardation-interacting protein 1 (NUFIP1) is transported from the nucleus to lysosomes and more importantly to autophagosomes stimulating the degradation and recycling of ribosomes, a process called ribophagy PMID: 29700228.

1)     Lines 271-281 the authors discuss DHODH inhibition in MYC driven cancers. Previously, the authors discuss how MYC primarily drives purine synthesisq, while DHODHi will inhibit pyrimidine synthesis only. It would be helpful if the authors could discuss here why inhibiting the pathway MYC does not turn on, actually slows the growth of these cancer cells. For example, would IMPHDi inhibitors actually work better?

Response 2: MYC does not only regulate purine synthesis, but also pyrimidine synthesis genes such CAD and CTPS (Satoh, K., et al., 2008). MYC is a master regulator of the expression of genes involved in the purine and pyrimidine synthesis pathways. Even if MYC gets activated in response to pyrimidine or purine depletion, proliferating cells cannot survive due to lack of building blocks required for cell division.

2)     With regard to reference 78 (lines 293-298), it may also be helpful to cite PMID: 30100185. This paper provides further evidence that ASS1 loss drives pyrimidine synthesis, and also has a nice discussion on nucleotide imbalance, which the authors later discuss.

Response 3: The reference PMID:30100185 was added.

3)     In lines 486-489 in discussing the use of IMPDH inhibitors in cancer, it may be worth discussing how the immunosuppressive effects of these drugs will interact with immunotherapies that are increasingly used clinically. How will this affect the ability to target nucleotide metabolism in cancer?

Response 4: This is an important point. We added few sentences and a reference about the immunosuppressive drugs and cancer therapy.

Line 505-516: Pharmacological inhibition of purine synthesis as a means to treat cancer has been examined for many years, with IMPDH considered as a possible target. Both MPA and mizoribine are already used in humans as immunosuppressants in organ transplantation and autoimmune diseases; mizoribine is particularly well tolerated [123-125]. Several epidemiological studies have consistently observed an increased risk of cancers in patients receiving organ transplant treated with immunosuppressive drugs PMID: 12581698. This raises the question of the role of these immunosuppressive drugs in the context of cancer immunotherapies. Cancer immunotherapy is concentrated on the immune system and is frequently more targeted than conventional cancer treatments such as chemotherapy or radiotherapy. The use of chemotherapy based on antimetabolite targeting nucleotide metabolism should certainly not be combined with immunotherapy which is supposedly used to activate the immune system. Further studies are required to estimate the actual cancer risk of the immunosuppressive drugs and notably of IMPDH inhibitors.

Reviewer 3 Report

This is a comprehensive and well-written review paper describing the signaling pathways that regulate de novo purine and pyrimidine synthesis. There are minor grammatical/stylistic errors throughout and this could benefit from the close read of an editor. The only content I would suggest adding is a summary of some of the historical studies performed with the DHODH inhibitor brequinar (see example here: https://link.springer.com/article/10.1023/A:1016066529642 ). These did not show efficacy as single agents or in combination, possibly because of compensation through salvage pathways.

Author Response

This is a comprehensive and well-written review paper describing the signaling pathways that regulate de novo purine and pyrimidine synthesis. There are minor grammatical/stylistic errors throughout and this could benefit from the close read of an editor. The only content I would suggest adding is a summary of some of the historical studies performed with the DHODH inhibitor brequinar (see example here: https://link.springer.com/article/10.1023/A:1016066529642 ). These did not show efficacy as single agents or in combination, possibly because of compensation through salvage pathways.

Response: We thank the reviewer for the very nice comments. We have addressed this comment by citing the manuscript suggested and adding one sentence about the lack of efficiency of brequinar in clinical trials.

Line 477-480: “Phase 1 clinical trials using brequinar did not show any clinical efficacy in cancer patients when this drug was used as a single agent or in combination with cisplatin PMID: 9740540, suggesting that compensation mechanisms exist and could be mediated by an increase in the activity of the pyrimidine salvage pathway.”

Reviewer 4 Report

The review is well written and very informative accompanied by comprehensive figures. A minor concern would be that I wish to see a bit more stretched discussions of each section. I think that the current version is already high quality and support the publication in Cancers.

Minor points:

-The word "long-term regulation" may be replaced with "slow regulation" or some other words if authors use "acute".

-The title would be "Cancer cells tune the signaling pathways to empower de novo synthesis of nucleotides".

- Italic is preferable for "de novo".

-(Line195) NADP should be NADP+.

-(Line340) diphosphate should be bisphosphate.

-mutant p53 has several meanings. Please indicate that this mutant is an activated-form.

Author Response

The review is well written and very informative accompanied by comprehensive figures. A minor concern would be that I wish to see a bit more stretched discussions of each section. I think that the current version is already high quality and support the publication in Cancers.

Minor points:

-The word "long-term regulation" may be replaced with "slow regulation" or some other words if authors use "acute".

Response 1: We thank the reviewer for the very nice comment. We have amended long-term regulation and replace it with slow regulation when acute was used in parallel.

-The title would be "Cancer cells tune the signaling pathways to empower de novo synthesis of nucleotides".

Response 2: Thank you for this suggestion. We have amended the title according to the reviewer’s suggestion.

- Italic is preferable for "de novo".

Response 3: We have converted de novo in italic.

-(Line195) NADP should be NADP+.

Response 4: This was amended.

-(Line340) diphosphate should be bisphosphate.

Response 5: Diphosphate is now changed to bisphosphate

-mutant p53 has several meanings. Please indicate that this mutant is an activated-form.

Response 6: Line 305-306: The “activated” term has been added to the text.
